# Preventing Childhood Obesity in Primary Schools: A Realist Review from UK Perspective

**DOI:** 10.3390/ijerph182413395

**Published:** 2021-12-20

**Authors:** Sharea Ijaz, James Nobles, Laura Johnson, Theresa Moore, Jelena Savović, Russell Jago

**Affiliations:** 1The National Institute for Health Research, Applied Research Collaboration West (NIHR ARC West), University Hospitals Bristol and Weston NHS Foundation Trust, Bristol BS1 2NT, UK; james.nobles@bristol.ac.uk (J.N.); theresa.moore@bristol.ac.uk (T.M.); jelena.savovic@bristol.ac.uk (J.S.); russ.jago@bristol.ac.uk (R.J.); 2Bristol Medical School, University of Bristol, Bristol BS8 2PS, UK; 3Centre for Exercise, Nutrition and Health Sciences, School for Policy Studies, University of Bristol, Bristol BS8 1TZ, UK; laura.johnson@bristol.ac.uk; 4Cochrane UK Methods Support Unit, Editorial & Methods Department, London SW1Y 4QX, UK

**Keywords:** childhood obesity, primary school, realist synthesis

## Abstract

Childhood obesity is a global public health concern. While evidence from a recent comprehensive Cochrane review indicates school-based interventions can prevent obesity, we still do not know how or for whom these work best. We aimed to identify the contextual and mechanistic factors associated with obesity prevention interventions implementable in primary schools. A realist synthesis following the Realist And Meta-narrative Evidence Syntheses–Evolving Standards (RAMESES) guidance was with eligible studies from the 2019 Cochrane review on interventions in primary schools. The initial programme theory was developed through expert consensus and stakeholder input and refined with data from included studies to produce a final programme theory including all of the context-mechanism-outcome configurations. We included 24 studies (71 documents) in our synthesis. We found that baseline standardised body mass index (BMIz) affects intervention mechanisms variably as a contextual factor. Girls, older children and those with higher parental education consistently benefitted more from school-based interventions. The key mechanisms associated with beneficial effect were sufficient intervention dose, environmental modification and the intervention components working together as a whole. Education alone was not associated with favourable outcomes. Future interventions should go beyond education and incorporate a sufficient dose to trigger change in BMIz. Contextual factors deserve consideration when commissioning interventions to avoid widening health inequalities.

## 1. Background

The world has witnessed a rapid increase in the prevalence of childhood obesity in the last three decades. A third of children in England are overweight or have obesity by the time they leave primary school [1]. Strategies to prevent excessive weight gain are therefore needed.

Obesity is now widely accepted as an outcome of a complex and obesogenic system [2,3,4]. Population levels of obesity are known to be the product of many interrelated and interdependent factors [5], and in response, researchers, practitioners and policy makers have started to call for the implementation of a systems approach. These approaches acknowledge that many different sectors, organisations, communities, families and individuals need to come together to systematically address the root causes of obesity [2]. Given that children spend approximately 25% of their waking hours in schools, and the important role that schools play within society, they serve as a key setting for obesity prevention efforts [6,7]. Although, schools cannot be expected to prevent childhood obesity on their own, they make up an important part of the system where interventions can go beyond targeting individual responsibility.

The latest Cochrane review [8] found that school-based obesity prevention interventions can achieve small changes in standardised body mass index (BMIz) over a school year. However, as interventions varied widely in the design and degree of success, the review does not highlight to public health professionals which intervention features work best, for whom and in what contexts. Realist reviews can help answer these questions by identifying contexts and mechanisms associated with intervention outcomes [9,10,11].

The aim of this realist review was to identify, and understand, the contextual and mechanistic factors associated with the outcome of school-based obesity prevention studies included in the Cochrane review of Brown et al. [8], which may be implemented within UK primary schools.

## 2. Methods

We carried out a realist review underpinned by the Realist And Meta-narrative Evidence Syntheses–Evolving Standards (RAMESES) guidance and the existing realist reviews in similar fields [10,11]. The study was registered with PROSPERO in July 2019 (CRD42019142192) [12].

### 2.1. Development of a Programme Theory

We developed an initial programme theory (Figure 1) using our team expertise in obesity prevention, and intervention development and evaluation.

Patient and Public Involvement: We sought external stakeholder opinion [13,14]—via an online consultation—to facilitate our understanding of the UK primary school contexts, and what stakeholders (school staff, management and organisations that work with primary schools) consider important for our review’s question.

The initial theory outlined the contextual and mechanistic factors that may be associated with a change in BMIz among children aged 4–12 years old exposed to a primary school-based intervention. This programme theory was further developed with stakeholder input and refined with data from included studies over the course of the review in an iterative manner. A Appendix A illustrates how the programme theory evolved.

### 2.2. Inclusion/Exclusion Criteria

Our sample frame was the recent Cochrane review (search period from database inceptions to June 2015) “Interventions for Preventing Obesity in Children” which included 153 studies [8]. We included studies which met the following criteria: conducted in primary schools; included children aged 4–12 years; interventions aimed to prevent obesity; and presented the mean BMIz as an outcome.

### 2.3. Data Extraction (Selection and Coding)

Two reviewers (S.I., J.N.) assessed all of the studies included in the Cochrane review to determine if a study met our inclusion criteria. The data were extracted into a standardised template (see Appendix A) which evolved as the review progressed. Whenever we identified a new context or mechanism during the data extraction, we added these to data extraction forms and then revisited the previously extracted studies to ensure data were not overlooked. Over repeated rounds, and along with input from topic experts on the team (J.N., L.J. and R.J.), we reached consensus over the coding for all of the extracted texts.

### 2.4. Rigour Assessment

We operationalised rigour assessment into a four-point scale based on the RAMESES definition of rigour [15] which are presented below. We employed risk of bias [16] judgements for the outcome as reported in the Cochrane review [8]. These decisions were made case by case and agreed between two reviewers (S.I., J.N.) (see example in Appendix A)

The four categories of rigour for studies were:Highly rigorous data (++): Arguments/data for the context-mechanism-outcomes (CMOs) are appropriate (underpinned with theory and data), and study was at a low risk of bias for our outcome.Rigorous data (+): Arguments/data presented are appropriate for CMOs, and study is not at a low risk of bias for our outcome.Unclear rigour of data (?): No or weak arguments/data presented for CMOs, irrespective of whether study is at a low risk of bias for our outcome.Data not rigorous (−): Contrary or unreliable arguments/data presented, irrespective of whether study is at a low risk of bias for our outcome.

### 2.5. Data Synthesis

Synthesis was a two-stage process. We first presented data on the CMO configurations at study level. Thus, producing a programme theory diagram for each study describing its CMO configurations. Then, for stage 2, we collated the CMO configurations from each study into a single, synthesised programme theory diagram (Figure 2).

We also summarised data reported on costs and sustainability of the interventions (Appendix A), as stakeholders considered these important.

### 2.6. Analysis of Subgroups or Subsets

We present programme theories for effective (defined as statistically significant BMIz change favouring intervention as seen in the Cochrane review) and ineffective interventions in Appendix A. We also synthesised studies with rigorous data alone to see any differences from main synthesis (see Appendix A).

## 3. Results

All of the 153 studies included in the Cochrane review were assessed at an abstract stage against our inclusion criteria. Of these, 29 studies met the criteria and were assessed in full texts (81 documents). Five studies (10 documents) were excluded at this stage as these were set entirely outside of the school [17,18,19] or did not involve primary school aged children [20,21]. Thus, 24 studies [22,23,24,25,26,27,28,29,30,31,32,33,34,35,36,37,38,39,40,41,42,43,44,45] (71 documents) were included in this realist review. See Appendix A for the study flow and lists of excluded and included study documents.

### 3.1. Included Study Characteristics

See details of the studies and extracted data in Table 1.

**Figure 2 ijerph-18-13395-f002:**
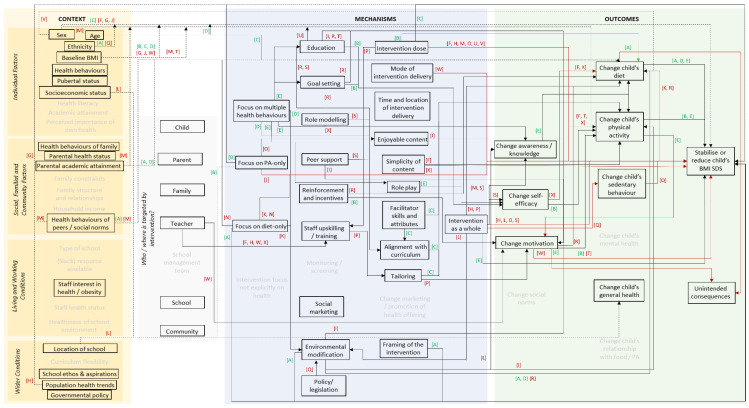
Final programme theory showing CMOs from all included studies. Dotted black lines indicate which contexts affected which outcomes. Continuous black lines from mechanism to outcomes indicate a favourable change (e.g., improved physical activity (PA) levels) while continuous red lines indicate lack of a favourable change (e.g., no difference in PA levels or an unfavourable change (e.g., lower PA levels). [] bracketed letters underneath the lines indicate respective studies for that CMO line. Green brackets refer to studies that found a favourable BMIz change (effective studies) and red refer to those that did not (ineffective studies): A = deRuyter, 2012 [24]; B = Khan, 2014 [32]; C = Li, 2010 [35]; D = Marcus, 2009 [36]; E = Spiegel, 2006 [44]; F = Fairclough, 2013 [25]; G = Cao, 2015 [22]; H = Sahota, 2001 [41]; I = Gutin, 2008 [28]; J = Lazaar, 2007 [34]; K = Damsgaard, 2014 [23]; L = Rush, 2012 [40]; M = Grydeland, 2014 [27]; N = James, 2004 [30]; O = Meng, 2013 [37]; P = Rosario, 2012 [39]; Q = Foster, 2008 [26]; R = Muckelbauer, 2010 [38]; S = Santos, 2014 [42]; T = Siegrist, 2013 [43]; U = Williamson, 2012 [45]; V = Herscovici, 2013 [29]; W = Johnston, 2013 [31]; X = Kipping, 2014 [33].

The majority of interventions addressed multiple health behaviours (16 studies), followed by diet alone (6 studies) and physical activity (PA) alone (3 studies). Interventions were most often tested in the USA (six studies), followed by the UK and China (three studies in each). Most (*n* = 16) interventions were delivered entirely during school hours and the majority of interventions (*n* = 13 studies) targeted children, their parents (or family) and teachers together. Teachers were the providers (deliverers) of interventions most often (18 studies) either exclusively (10 studies) or with a third party such as researchers, children, health or PA experts (8 studies). The interventions’ durations ranged from 3 months to 4 years with a median of 12 months (IQR 7.5 to 24).

### 3.2. The Final Programme Theory

Amendments to the programme theory throughout the review period can be seen from Figure 1 and Figure 2 and in Appendix A. Six new contexts (age, health behaviours of child, pubertal status, parental health status, parental academic attainment and population health trend) and six new mechanisms (social marketing, timing of intervention delivery, enjoyability of content, simplicity of content, role play and alignment with the curriculum) in total were added to the programme theory over five iterations (available in Appendix A). We found evidence on 16 contexts and 20 mechanisms from the 24 included studies. We present our findings below starting from most cited to least cited contexts and mechanisms across studies.

#### 3.2.1. Contextual Factors

**Baseline BMI classification** was a major contextual factor for intervention effect. Four studies found their interventions worked better for children with overweight or obesity in contrast to children of a healthy weight [22,31,34,35,36]. Two studies found their intervention worked only for children who were of a healthy weight at baseline [27,32]. Only one study discretely tailored the intervention differently for the two groups so as to minimize “potential for stigmatizing overweight kids” [45], albeit with no effect difference in BMIz.

**Sex** appeared to be the next noteworthy context. Girls were reported on several occasions to benefit more from interventions in terms of favourable BMIz, PA or diet change [22,25,27,29,34,35]. Study authors argued that girls may be more concerned about their body image and weight, therefore, more likely to adhere to the educational content of the interventions. Compared to boys, girls also maintained changes in BMIz after the interventions stopped [25,35].

For **ethnicity**, one study [26] found evidence that black children benefited more from their intervention than white children. Conversely, another [38] argued that, since the educational component of the intervention was not tailored to account for cultural differences, their intervention may have been less effective for migrant (non-German) children, although no effect difference by this variable was seen. Two studies [28,40] tailored their intervention content for cultural differences and found no difference in the outcomes between children of different ethnicities.

Older **age** children achieved lower BMIz [36] and higher PA levels [35]; Li et al. [35] argued this may be because older children are better able to understand and follow the directions associated with the intervention.

**Parental academic attainment** also impacted an intervention’s effects. In two studies, the children of parents with lower academic attainment were less likely to make dietary changes [24,36]. These children were also less likely to complete the intervention [23,24].

**Peer behaviour and social norms** were noted contexts in two studies [24,27]. DeRuyter et al. [24], who replaced children’s sugary drinks with artificially sweetened ones, noted that the social norm among Dutch children to bring a sugar sweetened drink with them to school allowed for easy switch to an artificially sweetened drink. So, the intervention is unlikely to work in countries where sugar-sweetened drinks are not routinely consumed at school. Grydland et al. [27], who offered fruit and vegetable snacks at break time, noted that fruit, but not vegetable, intake increased amongst the children. They argued that this was because in Norway, vegetables are often eaten during evening meal, which is why only fruit consumption increased.

**Population health trends** appeared to affect how an intervention worked in one study where the population prevalence of childhood overweight and is high, it is unlikely that a simple educational intervention will suffice [41]. Other contexts potentially influencing an intervention’s effect on a child’s health were **good parental health status**, [22] rural **location of school** [40] and **high socioeconomic status** (SES) [40].

#### 3.2.2. Mechanisms

**Education** was the most used mechanism (18 studies). Education alone led to a change in *motivation* in three studies [28,38,43] and to a change in *self-efficacy* in one [32], but not BMIz. Spiegel et al. [44] demonstrated that education, when delivered through mechanisms of **goal setting**, **role play** and **tailoring**, would change *knowledge*, *self-efficacy* and *motivation*. The knowledge change was argued to have brought about change in a child’s *diet, PA levels and BMIz*. Williamson et al. [45] provided evidence that education combined with **alignment with the curriculum** as a mechanism could change a child’s PA.

The second most cited mechanism was sufficient **intervention dose.** Three studies argued that a sufficient intervention dose brought about a significant *BMIz change* [32,35,36]. Ten [35] and thirty [36] minutes of integrated daily PA over 12 and 48 months, respectively, was effective in changing BMIz for children with overweight or obesity. While 70 min of intermittent moderate to vigorous physical activity physical activity (MVPA), five times a week, for nine months was argued as sufficient to change the BMIz in children with healthy weight at baseline [32].

Several other studies argued that the intervention dose was too low to achieve a BMIz reduction [25,27,29,33,37,41,43,45]. However, most of these involved educational health promotions and little enabling of PA. For example, 20 months [27] and 28 months of PA promotion in school [45] was insufficient to alter BMIz compared with the control group. While BMIz stayed unaffected, the children’s *PA levels* improved after 3 years of 80 min MVPA at least twice a week [28] but not after 6 months of 10 min daily MVPA [37]. Both interventions claimed to be enjoyable (i.e., an additional mechanism).

Insufficient intervention dose was also proposed as a reason for unchanged *diet behaviour* [23] because the intervention could only influence food consumed within school hours, and therefore had limited potential to change total daily intake. Kipping et al. [33] hypothesised **self-efficacy** as a mechanism for change in diet and activity behaviours, however, Kipping et al. suggested that intervention dose was not enough to change self-efficacy. They also suggested that change in PA requires more intense PA interventions, however, it was also noted that given how busy schools and staff already are it may not be feasible.

**Environmental modification** often altered food options available for children but this was not always associated with change in dietary behaviour [22,23,26,29,40,41,43,45]. Only in two studies [36,38] was environmental modification associated with a change in child’s diet, and with a BMIz change in one [36]. These modifications consisted of: (a) modifying the arrangement of school lunches in self-service areas: fruit and vegetables were placed before other options [36] and (b) the installation of water fountains in school premises [38]. The authors argued that these environmental modifications–once implemented–led to sustainable changes in dietary behaviours.

Two studies used environmental modification as a mechanism to bring about change in the children’s PA levels [28,43]. Gutin et al. [28] created what they termed a “fitogenic environment” through the provision of additional PA afterschool, whilst Siegrist et al. [43] made modifications to the classrooms, halls and playgrounds to encourage PA. Both studies demonstrated positive impacts on PA levels, but not on BMIz.

**Intervention as a whole** was cited as a mechanism in six studies. We assume that most interventions are designed to work as a whole, however, in the context of this realist review, only a small number of studies were explicit in stating that it was the entirety of the intervention that brought about a change in an outcome, with one of these achieving *BMIz change* [44]. Spiegel et al. [44] attributed the BMIz change to the various intervention components (via **role play, goal setting, tailoring and alignment with the curriculum**) working “in concert … creating something greater than the sum of the parts.” Two other studies, Sahota et al. [41] and Rosario et al. [39], reported that the intervention as a whole only changed *dietary intake*. Similarly, Foster et al. [26] found that their intervention, as a whole, only led to a change in *sedentary behaviour*, with Grydland et al. [27] Rush et al. [40] and Santos et al. [42] citing their interventions as a whole changed *knowledge and awareness* of health behaviours.

**Alignment with the curriculum** and **staff upskilling/training** were often employed together [22,25,26,31,35,41,42,43] aiming to **educate** the children in order to change the behaviour and yet led to behaviour change in only one study [35]. This was achieved via additional contributions from **tailoring** of this intervention to the age group and an optimal intervention dose.

**Tailoring** was employed in four studies [35,39,44,45] and, as mentioned above, only in one [35], it led to the desired behaviour change in children. Tailoring was demonstrated via age- and space-appropriate exercises where students and teachers were allowed to develop new activities in one study [35], options to increase intensity of aerobic exercises in class in another [44] and a software programme recognizing children with overweight and offering them different content in one [45]. The fourth study [39] ensured that intervention content could be tailored by the teachers themselves in order to best serve the needs of their pupils.

Five studies reported their interventions to have **enjoyable content** [28,34,37,39,41]. However, only one of these studies [28] highlighted that their enjoyable PA content (by offering different activities and enabling children to see their progression) changed motivation.

**Simplicity** of the intervention and/or intervention content was cited in three studies [25,30,33], all from the UK. One argued that their simple message led to change in child’s PA levels [25]. The other two studies [30,33] found their simple interventions not successful as a mechanism in changing BMIz. It must be reiterated here that we took the authors’ labelling of their intervention as “simple” and there is limited interpretation possible from them. Kipping et al. [33] employed child education, role modelling, teacher training and parent counselling. They argue in their conclusions that such “simple school-based interventions that are designed to minimise costs” cannot bring about major change in diet and PA. Fairclough et al. [25], on the other hand, although employed education and training for the child, teacher and parent, focussed on changing the curriculum to include the simple message ‘move more sit less’ which they believe was a simple non-prescriptive approach.

### 3.3. Gaps in Evidence

We found no evidence for some individual contextual factors (such as a child’s academic attainment, health literacy, perceived health status and perceived importance of own health), and some family factors (family constraints, family structures and relationships and household income). Moreover, missing was evidence on the type of school (public or private), slack (resource) available in school, staff health status, healthiness of the school environment and curriculum flexibility. The mechanisms not addressed in any studies were monitoring/screening, change marketing/promotion of health offering and changing social norms.

### 3.4. Reporting of Costs

Eight studies reported cost or resource use (see Appendix A). Costs for these varied interventions in current GBP values could range from GBP 12 [37] to over GBP 1300 [32] per child per year.

### 3.5. Reporting on Sustainability of Intervention

Eleven studies highlighted intervention features which they believe increased the sustainability of the intervention—see Appendix A. These were: stakeholder involvement in intervention design and development, delivering it within the existing resources of the school, collaborating with the relevant authorities and sectors and adaptable (flexible) intervention content.

### 3.6. Findings of Sensitivity Analysis

Restricting our analysis to only rigorously conducted studies (*n* = 11; judged either **++** or **+**) [23,24,28,30,31,32,33,34,35,36,42], we found the key contexts of influence were still baseline BMI [31,32,34,35,36], parental educational attainment [24,36] and sex [34,35] (see Appendix A). Among the mechanisms, intervention dose [23,28,32,33,35] stood out again along with environmental modification [23,24,28,36] as the most often cited.

## 4. Discussion

This realist synthesis found that female sex, and older age, alongside higher parental academic attainment, are key contexts for intervention effectiveness. While some interventions benefited children with a higher baseline BMIz status, others benefited already healthy weight children. Girls appeared to benefit from the interventions due to the influence of social norms surrounding body image, which is in line with the findings of a recent large-scale study in the UK [46]. Future studies should therefore consider how interventions may better meet the needs of boys while also addressing the negative social norms surrounding female body image. Similarly, interventions should ensure that they are not just effective for children of highly educated parents, or those without overweight and obesity, because this may inadvertently widen health inequalities.

Despite socioeconomic status (SES) being a well-known moderator of intervention effect for health promotion interventions [47], it was formally explored in only one included study [40]. This limited evidence on SES was also reported in a recent overview of obesity prevention in adolescents [48]. Thus, it is important to consider here how interventions may widen health inequalities if they offer more favourable outcomes for people who are socioeconomically better off. As aforementioned, parental education, which is a proxy indicator for SES [49,50], was associated with intervention uptake and effect. Educational attainment is only one domain associated with SES, and so future studies should separate the effects of SES from parental education levels. This will allow us to target the context that is preventing the intended mechanisms from working.

The perceived sufficiency of the intervention dose appeared to affect BMIz in various contexts. However, what constituted sufficient or optimal dose (or dose range) was not specified. Dose can include frequency and duration of an intervention session (per week or per month) as well as the duration of the entire intervention (in months or years). Which, if any, or what combination of these components may be more beneficial is unknown. A recent systematic review found no link between dose and weight outcomes, which they argued could be either because behaviour change is non-linear or because of the varied reporting of dose [51]. Given the emphasis placed on intervention dose by many studies in this review, this is a key area for future clarification.

Interventions adopting environmental modification require little individual agency to alter health behaviours, and therefore may be simpler and more sustainable than educational interventions [52]. However, the limited evidence on changing BMIz is important as it may suggest further intervention is required to impact health beyond behaviour change. The simplicity and enjoyability of an intervention were argued to have the potential to change the activity and diet related health behaviours. However, we need clarity on what children deem simple or enjoyable.

Interventions using education as the sole mechanism appeared to have a limited impact on behaviour or BMIz. This aligns with the broader evidence base, which suggests educational interventions are unlikely to elicit effective changes for children [53,54], and for the general population [55]. Relying on individual agency is unlikely to translate into substantial or sustained behavioural change, and consequently obesity prevention [56].

### 4.1. Comparison with Existing Literature

There is no shortage of evidence syntheses of childhood obesity preventive interventions: a recent overview included 66 meta-analyses and systematic reviews on the topic [57]. Syntheses usually find that interventions addressing diet and PA are more promising than targeting either behaviour alone. However, the high heterogeneity across the studies provided the rationale for our realist synthesis, which aimed to understand the underlying contextual and mechanistic factors that help interventions generate outcomes.

Our findings broadly align with recent realist reviews in the area of childhood PA [10,11]. These reviews found that sex (contextual factor) and goal setting, tailoring and intervention dose (mechanistic factors) were linked to the intervention outcomes. Tailoring seldom arose within our review, perhaps due to different operational definitions for what tailoring constitutes or due to the contextual differences between study settings and populations; the review of Hnatiuk et al. [10] focussed on children aged 0–5 in pre-school settings, whilst the review of Brown et al. [11] looked at family-based interventions for children of primary school age (5–12 years). There may be more scope to tailor interventions within these settings in contrast to a primary school setting. While many interventions aimed to align or embed content within the school curriculum, they rarely hypothesised this mechanism to affect BMIz. It may also be that processes were not in place to measure these mechanisms in studies and is not a sign per se that these are ineffective. It would be good in the future to consider a priori how mechanisms would act together to bring about a change and evaluate if the process happened as anticipated.

### 4.2. Strengths and Limitations of Our Realist Synthesis

The key strength of this review is that we approached the existing evidence on obesity prevention to understand *why* and *how* an intervention works rather than *whether* it works. The realist synthesis—a relatively new method—allowed us to address these questions which are important to decision makers. We present new insights into the evidence beyond a traditional meta-analysis on the intervention outcome and avenues for future exploration. The findings should help implement an effective obesity prevention intervention in practice.

The review included a large, robust dataset from the most recent Cochrane review [8]. We included all of the qualitative and process evaluations from the 24 studies, amounting to 71 documents in total. This led to rich data for analysing CMO configurations. We restricted our sampling frame to the Cochrane review, which is up to date until 2015, so we may have missed new interventions, contexts or mechanisms, which is a limitation. The planned Cochrane update effort has identified (but not extracted) a further 162 relevant trials published between 2015 and 2018 and the search for trials after 2018 is ongoing. However, the included interventions in the Cochrane review did not change substantially since its first publication in 2002 (i.e., with a downstream focus on individual behaviour change) [8] and this was confirmed in a recent secondary analysis of the Cochrane review [58] using a wider determinants of health lens. The findings indicate that (a) the majority of studies target individual dietary and PA behaviours, and (b) the focus of childhood obesity prevention interventions has not changed over time since 1993—the publication date of the oldest study included in the Cochrane review.

This is a limitation of the evidence base, whereby the focus is traditionally on behavioural change at individual levels, and environmental or policy interventions targeting the wider determinants of health (upstream) are rarely evaluated in randomised trials [7,48]. Policy interventions can be evaluated using randomised designs [59], where one geographical or political region may implement the policy sooner than others (waitlist control or stepped wedge design). Where randomisation is not feasible, interrupted time series or controlled before and after designs could be employed to evaluate wider determinants of health and policy interventions [60]. That said, two recent systematic reviews [7,61] of natural experiment studies also found that the included studies predominantly focussed on downstream determinants of childhood overweight and obesity. Thus, we anticipate it is unlikely that the focus of interventions has changed dramatically between 2015 and 2020.

### 4.3. Implications for UK-Based Primary Schools

The stakeholder consultation indicated that UK primary schools have limited resources to take on obesity prevention tasks. With no evidence in the review to support the usefulness of additional health education for changing BMIz, it may be difficult to justify teachers doing this. Education may be important but is insufficient on its own to change BMIz. Implementing an environmental modification (such as the installation of water fountains, changed canteen offerings) may be perceived more favourably by school staff. This may also bypass the reliance on individual agency for behaviour change. One suggestion [62] to optimise implementation of a school intervention is to involve delivery staff (school staff, management or third party) in the design and development of the intervention. We recommend including children in this planning.

Given the limitation of school finances in the UK, cost is a major consideration for any intervention. Whilst obesity prevention interventions are likely to be cost effective in the long-term [63], these returns may not be seen by the education sector (or individual schools), and thus the immediate investment required to establish a new initiative may be negatively perceived by the stakeholders. Unfortunately, there was insufficient information in the studies to analyse the costs of different intervention types. We need full cost reporting for future interventions, including a breakdown of the costs per intervention component, to facilitate decision making.

## 5. Conclusions

Our findings indicate that being female and older, and having parents with a high academic attainment can help children benefit from obesity preventive interventions, while baseline BMI can affect intervention outcomes variably. The potential ramifications for health inequalities with these contexts must be kept in mind by both commissioners and researchers. Sufficient intervention dose and environmental modifications in schools are mechanisms that may help achieve the desired outcomes. In addition, an intervention that worked as a whole rather than a collection of separate components can better achieve the desired outcome, illustrating the interdependent nature of the intervention mechanics—the effect being greater than the sum of its parts. That said, few mechanisms favourably influenced BMIz, and were more likely to only change knowledge, motivation and some health behaviours.

## Figures and Tables

**Figure 1 ijerph-18-13395-f001:**
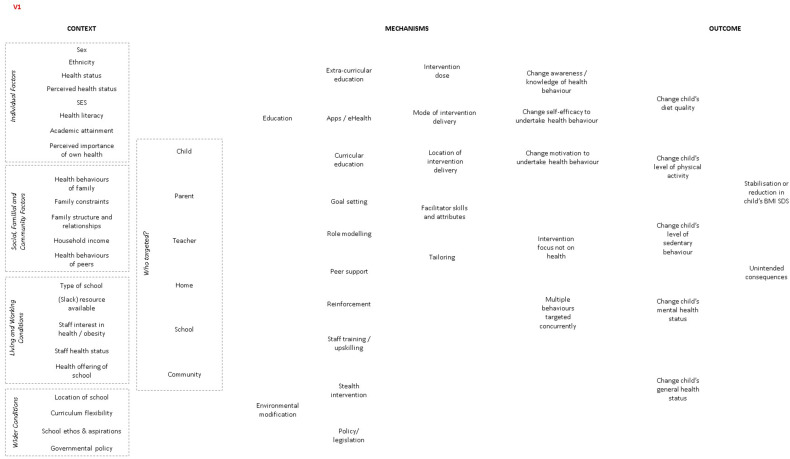
The initial programme theory.

**Table 1 ijerph-18-13395-t001:** Characteristics of included studies.

Study and Location	Intervention Content and Delivery	Contexts Identified	Mechanisms Identified	CMO Configurations	Rigour
**Effective studies**					
de Ruyter, 2012 [24] The Netherlands	**Description:** Double blind RCT, replacing sugary drinks (regularly consumed in school breaks and at home) with identical tasting sugar free drinks. **Provider:** Third party (researchers) **Timing:** N/A–drinks available at home and school; 18 months duration. **Target group:** Children; parents; teachers; school.	Age Ethnicity Health behaviours Health status Health behaviours of peers/social norms Parental academic attainment	Focus on diet only Environmental modification Reinforcement and incentives Mode of delivery Time and location of delivery Framing of intervention	Parental academic attainment change diet Ethnicity change BMIz Focus on diet alone → Change in child’s BMIz	**++**
Khan 2014 [32] USA	**Description:** Two hours of daily PA, five days/week for nine months; 15 min of education and healthy snack. **Provider:** Third party (undergrads, researchers). **Timing:** After school hours; nine months duration. **Target group:** Children.	Age Pubertal status Health status	Focus on PA only Intervention dose Education Goal setting Reinforcements and incentives Facilitator skills and attributes Changing self-efficacy Changing motivation	Healthy weight → Change BMIz Focus on PA alone → Change BMIz	**++**
Li, 2010 [35] China	**Description:** Two daily 10 min MVPA sessions conducted in the break between classes with variety of safe, moderate, age- and space-appropriate activities. **Provider:** Teacher. **Timing:** During school hours; 12 months duration. **Target group:** Children.	Sex Age Health status Location of school	Focus on PA only Intervention dose Education Role modelling Change awareness/knowledge Reinforcements and incentives Alignment with curriculum Tailoring Facilitator skills and attributes	Sex + baseline BMI → change BMIz Staff training → Facilitator skills and attributes → Change awareness/knowledge Intervention dose → Change BMIz Facilitator skills + tailoring + alignment with curriculum → Change BMIz	**+**
Marcus, 2009 [36] Sweden	**Description:** 30 min of daily PA was integrated into the curriculum. School lunch and afternoon snack were made healthier by adding fruit and vegetables. Awareness raising intervention provided for staff and parents. **Provider:** Teacher. **Timing:** During school hours; 48 months duration. **Target group:** Children, parents school staff.	Ethnicity SES Age Health status Parental education attainment	Focus on multiple behaviours Alignment with curriculum Environmental modification Change knowledge/awareness	Focus on multiple behaviours + child → Environmental modification → change in child’s diet → Change child’s BMIz Focus on multiple behaviours → Alignment with curriculum	**+**
Spiegel 2006, [44] USA	**Description:** Seven modules of educational content for children. Modules on (1) general wellness, (2) reflective self-analysis, (3) principles of PA, (4) principles of diet and nutrition, (5) learning about the body, (6) genetics and family health and (7) practical application of acquired knowledge. Ten mins of PA each day during class time. **Provider:** Teacher. **Timing:** During school hours; nine months duration **Target group:** Children, family, teacher.	None identified	Focus on multiple behaviours Intervention as a whole Education Goal setting Role play Tailoring Alignment with curriculum Change knowledge and awareness Change self-efficacy Change motivation	Focus on multiple behaviours + role play → change self-efficacy Focus on multiple behaviours → change knowledge and awareness → change child’s diet + PA → change in BMIz Change motivation → unintended consequences (academic improvement) Intervention as a whole → change in BMIz	**?**
**Ineffective studies**					
Fairclough, 2013 [25] UK	**Description:** One hour of content per week over 20 weeks. Intervention provided teachers with lesson plans, worksheets, homework tasks, lesson resources and a CD-ROM. Topics covered PA and diet, and aligned with the UK Healthy Schools programme. Developed with parents, children and teachers input. **Provider:** Teacher. **Timing:** During and after school hours; five months duration **Target group:** Children, family, teacher.	Sex Ethnicity SES	Focus on multiple behaviour Education Staff upskilling and training Intervention dose Simplicity of content Alignment with curriculum Framing of intervention	Sex → change BMIz Simplicity of content → Child’s PA Intervention dose ↛ change child’s diet + PA Intervention dose ↛↛ change BMIz	**?**
Cao, 2015 [22] China	**Description:** Six hours of health educational content per semester. Intervention also includes regular newspapers, brochures, seminars, and morning meetings. Offer one hour of PA per school day. Lower fat content and more fruits and vegetables available at canteens. **Provider:** Teacher, parent. **Timing:** During and after school hours; 34 months duration **Target group:** Children, parent, teacher	Sex Health status Parental health status Location of school (urban China)	Focus on multiple behaviours Education Peer support Staff upskilling and training Environmental modification Facilitator skills and attributes Alignment with curriculum	Sex → change BMIz Health status → change BMIz Parental health status → change BMIz	**?**
Sahota, 2001 [41] UK	**Description:** Teacher training, modifications of school meals and the development and implementation of school action plans designed to promote healthy eating and PA over one academic year. Developed with parent, teacher, and child input. **Provider:** Teacher, school. **Timing:** During school hours; nine months. **Target group:** Children, teacher, school.	Population health trends (secular trends)	Focus on multiple behaviours Education Environmental modification Staff upskilling and training Intervention dose Enjoyable content Facilitator skills and attributes Alignment with curriculum Intervention as a whole	Population health trend → change BMIz Focus on multiple behaviours → enjoyable content Focus on multiple behaviours + enjoyable content Intervention as whole → change child’s diet Intervention as a whole ↛↛ change BMIz Intervention dose ↛ change BMIz	**?**
Gutin, 2008 [28] USA	**Description:** 40-min session of academic enrichment activities, followed by 80 min MVPA. Offered each day after school. Healthy snacks provided during break. **Provider:** Teacher. **Timing:** After school hours; 36 months duration. **Target group:** Children, teacher.	Sex Ethnicity SES	Focus on multiple behaviours Education Peer support Staff upskilling and training Environmental modification Intervention dose Time and location of intervention delivery Enjoyable content Change motivation	Sex or Ethnicity ↛↛ BMIz Education → change motivation Enjoyable content + Peer support → change motivation Environmental modification → change motivation Change motivation → change child’s diet Environmental modification → change child’s diet Intervention dose → change child’s PA	**+**
Lazaar, 2007 [34] France	**Description:** Two sessions of school PE per week (one hour per session). The which intensity and duration off sessions increased throughout the study with the aim that the 45 min of exercise in one hour is playful. **Provider:** Third party (state PE undergrads). **Timing:** During school hours; six months duration. **Target group:** Children.	Sex Health status	Focus on PA Peer support Enjoyable content Change knowledge and awareness	Sex → change BMIz Health status → BMIz change Focus on PA alone ↛ change BMIz	**?**
Damsgaard, 2014 [23] Denmark	**Description:** School lunch and snacks based on the New Nordic Diet, designed to cover 40–45% of the children’s daily energy intake (mid-morning snack, ad-libitum hot lunch, afternoon snack, fresh fruit or fruit-based dessert). Seasonal menus developed. Children participated in the cooking. **Provider:** Kitchen staff, school. **Timing:** Three months duration. **Target group:** Children, school.	SES Parental academic attainment	Focus on diet Environmental modification Intervention dose	Focus on diet → environmental modification Focus on diet ↛↛ change BMIz Intervention dose ↛↛ change child’s diet ↛ change BMIz	**+**
Rush, 2012 [40] New Zealand	**Description:** Project staff allocated to schools to model classes around various physical activities. Study also promoted active transport, lunchtime games, bike days and training for students to be leaders of PA. Project staff assisted school with healthy eating initiatives (e.g., canteen makeovers). Nutritional information included in weekly school newsletter. Parents asked to attend three information sessions and a 45-min practical nutrition class. Project staff helped teachers, parents and the local community via a range of activities (open days, edible gardens). **Provider:** Third party (project staff), teacher. **Timing:** During school hours; 24 months duration. **Target group:** Children, parent, teacher, school, community.	Ethnicity SES Location of school (urban/rural)	Focus on multiple behaviours Education Reinforcements and incentives Environmental modification Facilitator skills and attributes Alignment with curriculum Intervention as a whole Change awareness and knowledge	SES → change general health Location of school → change general health Intervention as a whole → Environmental modification → change awareness and knowledge Intervention as a whole ↛↛ change BMIz	**?**
Grydeland, 2014 [27] Norway	**Description:** Classroom-based dietary education using personally tailored computer software. Intervention also offered fruit/vegetable and PA breaks during day. Inspirational PA courses for teachers, and fact sheets to parents. Environmental component included active transport campaigns, PA equipment and suggestions for playground improvements. **Provider:** Teacher. **Timing:** During and after school hours; 20 months duration. **Target group:** Children, teacher, parent.	Sex Ethnicity SES Health status Health behaviours Health behaviours of peers/social norms	Focus on multiple behaviours Intervention dose Intervention as a whole Change awareness and knowledge	Sex → change child’s PA, Health status → change child’s PA Parental academic attainment → change child’s PA Health behaviours of peers/social norms → change child’s diet + change BMIz Intervention dose ↛ change BMIz Intervention as a whole → change (parental) awareness and knowledge	**?**
James, 2004 [30] UK	**Description:** Four educational components delivered to children by project staff: (1) a one-hour session delivered once per term on the balance of good health and promotion of drinking water, (2,3) one off sessions to create a rap/song about healthy diet and (4) a presentation and quiz. **Provider:** Third party (project staff), teacher. **Timing:** During school hours; 12 months duration. **Target group:** Children.	None identified	Focus on diet Education Simplicity of content Facilitator skills and attributes	Focus on diet ↛ change BMIz Focus on diet → change child’s diet Simplicity of content ↛ change BMIz	**+**
Meng, 2013 [37] China	**Description:** Classroom-based 10-min MVPA led by teachers. Sessions on nutrition and health six times for students (monthly), twice for parents and four times for teachers and health workers. **Provider:** Teacher **Timing:** During school hours; six months duration. **Target group:** Children, parent teacher.	Location of schools (urban China)	Focus on multiple behaviours Focus on diet Focus on PA Education Enjoyable content Intervention dose Intervention as a whole	Intervention dose ↛ change in BMIz Intervention as a whole →change in BMIz Focus on PA → enjoyable content	**?**
Rosario, 2012 [39] Portugal	**Description:** 12 nutritional education sessions of three hours each duration for children plus six month of teacher training. **Provider:** Teacher. **Timing:** During school hours; six months duration. **Target group:** Children, teacher.	Parental academic attainment Location of school (urban)	Focus on diet Education Staff upskilling and training Facilitator skills and attributes Tailoring Enjoyable content Intervention dose Intervention as a whole Change motivation	Teacher upskilling → tailoring → change motivation Intervention as a whole → change in child’s diet Intervention dose of teacher training → tailoring → motivation	**-**
Foster, 2008 [26] USA	**Description:** The School Nutrition Policy Initiative included: school self-assessment; nutritional education for parent, child and teacher; nutrition policy; social marketing campaign targeted at children; and parent outreach work via nutrition educators. **Provider:** Teacher, third party (nutrition educators). **Timing:** During and after school hours; 24 months duration. **Target group:** Children, parent, teacher, school.	Ethnicity SES Location of school Population health trend	Focus on multiple behaviours Education Reinforcement and incentives Staff upskilling and training Facilitator skills and attributes Tailoring Social marketing Environmental modification Policy/legislation Alignment with curriculum Intervention as a whole	Ethnicity → change BMIz Tailoring ↛ change BMIz Intervention as a whole → change child’s sedentary behaviour changes BMIz ↛ change BMIz	**-**
Muckelbauer, 2010 [38] Germany	**Description:** Combined environmental and educational intervention promoting water consumption: water fountains installed in schools, provision of reusable water bottles and lessons importance of water consumption **Provider:** Teacher, school. **Timing:** During school hours; 12 months duration. **Target group:** Children, teacher, school.	Ethnicity Health status Health behaviours of family Health behaviours of peers/social norms Health offering of school	Focus on diet Education Goal setting Reinforcement and incentives Environmental modification Alignment with curriculum Change motivation	Education + goal setting → change motivation → change child’s diet Reinforcement and incentive → change child’s diet Environmental modification → change child’s diet Change child’s diet ↛ change in BMIz	**-**
Santos, 2014 [42] Canada	**Description:** Older students received a weekly 45-min healthy living lesson from teachers (given training for two days). Older students acted as peer mentors, teaching a 30-min lesson to younger “buddies.” Two 30-min structured aerobic fitness sessions per week with student pairs. **Provider:** Teacher, child. **Timing:** During school hours; 10 months duration. **Target group:** Children.	Age Health status	Focus on multiple behaviours Education Role modelling Peer support Staff upskilling/training Alignment with curriculum Intervention as a whole Change awareness/knowledge Change self-efficacy	Intervention as a whole → change in awareness/knowledge Intervention as a whole → change self-efficacy Intervention as a whole ↛ change BMIz	**++**
Siegrist, 2013 [43] Germany	**Description:** 45 min per month of additional PE during school hours. Re-arrangement of the classrooms, halls and playgrounds to promote more PA. Worksheets, assignments and newsletters sent home to support PA. Measures to improve the quality of food sold at school snack bars. Parents provided with three hours of training, and teachers given nine hours. **Provider:** Teacher. **Timing:** During and after school hours; 12 months duration. **Target group:** Children, parents, teachers.	Health status	Focus on multiple behaviours Education Staff upskilling/training Environmental modification Alignment with curriculum Intervention dose Change motivation	Health status → change child’s PA Education → change motivation → change child’s PA Intervention dose ↛ change child’s PA	-
Williamson, 2012 [45] USA	**Description:** Environmental modification of school setting: (1) cues related to healthy eating and activity, (2) cafeteria food service and (3) PE programs. Behavioural modification: (1) educational program delivered as a part of class work, with synchronous on-line counselling and asynchronous email communications for children and parents. Teachers trained prior to, and throughout, the trial duration. **Provider:** Teacher. **Timing:** During school hours; 28 months duration **Target group:** Children	Sex Ethnicity Health status SES Location or schools (rural)	Focus on multiple behaviours Education Environmental modification Alignment with curriculum Tailoring Intervention dose	Health status → tailoring Tailoring ↛ change BMIz Education → change child’s PA Intervention dose ↛ change BMIz	**?**
Herscovici, 2013 [29] Argentina	**Description:** Four workshops (40 min each, once a month) on diet and PA (three for children and one for parents). Modifications made to school cafeteria menu. **Provider:** Third party (interdisciplinary team). **Timing:** During school hours; six months duration. **Target group:** Children, parents.	Sex SES Health behaviours of peers/social norms	Focus on multiple behaviours Education Environmental modification Facilitator skills and attributes Change awareness and knowledge Change motivation	Sex → change child’s diet Intervention dose ↛ change BMIz	**?**
Johnston, 2013 [31] USA	**Description:** Trained health professionals visited school three times per week to meet staff and provide suggestions for how to improve health messages across school. They trained and assisted teachers (60 h training and 40 h of supervised practice) to implement healthy messages in curriculum. They also helped to improve availability of nutrient-rich food at school cafeteria. **Provider:** Third party (trained health professionals), teacher. **Timing:** During school hours; 24 months duration. **Target group:** Children, parents, teachers, school.	Health status	Focus on multiple behaviours Reinforcements and incentives Staff upskilling/training Facilitator skills and attributes Mode of intervention delivery Alignment with curriculum Change motivation	Health status → change BMIz Staff upskilling/training + facilitator skills and attributes → teacher motivation Mode of intervention delivery → change BMIz Change motivation ↛ change BMIz	**+**
Kipping, 2014 [33] UK	**Description:** Training for teachers and teaching assistants provided by the study team. Teachers provided with 16 lesson-plans and teaching materials. Schools also provided with information that they could use in newsletters about the importance of PA, sedentary behaviour and diet. Parents provided with 10 parent–child interaction homework activities, and information on how to encourage their child’s health behaviours. **Provider:** Teacher, third party (multidisciplinary). **Timing:** During and after school hours; eight months duration. **Target group:** Children, parent, teacher, school.	Staff interest in health/obesity School ethos and inspirations Government policy	Focus on multiple behaviours Role modelling, staff upskilling/training Intervention dose Simplicity of content Change awareness and knowledge Change self-efficacy	Role modelling → change awareness and knowledge Intervention dose ↛ change child’s PA Intervention dose ↛ Change self-efficacy ↛ change child’s diet/PA Simplicity of content ↛ change BMIz	**++**

BMIz: standardized body mass index; CMO: context-mechanism-outcome; min: minutes; MVPA: moderate to vigorous physical activity; PA: physical activity; PE: physical education; RCT: randomised controlled trial; SES: socioeconomic status. ++: Highly rigorous data. +: Rigorous data? Unclear rigour of data: Data not rigorous →: context or mechanism produced a favourable effect ↛: context or mechanism did not produce favourable effect.

## Data Availability

All data included in this review is available as published research reports.

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
