# Peer review of "Preventing Childhood Obesity in Primary Schools: A Realist Review from UK Perspective"

_ijerph, 2021, doi:10.3390/ijerph182413395_

Round 1
Reviewer 1 Report
1. Title: Needs to insert ‘in’ between ~childhood obesity and primary schools.
2. Abstract: Please double check the term ‘REMESES’ ? -> needs to be corrected
3. Methods: It was easy to understand with clear and simple statements.
4. Results: The purpose of this study was systematically reflected with the contextual and mechanistic factors related to the outcome of school-based obesity prevention studies based on the Cochrane review.
5. Overall, this study is well organized and logically described. Especially, review processes on children’s obesity in primary schools have been shown in Table 1, indicating how the work is progressed with big efforts.
Author Response
Reviewer 1
- Title: Needs to insert ‘in’ between ~childhood obesity and primary schools.
Response: Thank you for pointing this out. We have corrected the error now.
- Abstract: Please double check the term ‘REMESES’ ? -> needs to be corrected
Response: Thank you for pointing this out. We have corrected the error now.
- Methods: It was easy to understand with clear and simple statements.
Response: Thank you
- Results: The purpose of this study was systematically reflected with the contextual and mechanistic factors related to the outcome of school-based obesity prevention studies based on the Cochrane review.
Response: Thank you
- Overall, this study is well organized and logically described. Especially, review processes on children’s obesity in primary schools have been shown in Table 1, indicating how the work is progressed with big efforts.
Response: Thank you for your appreciation of our work.
Reviewer 2 Report
My comments, which should be addressed before publication could be possible, are as follows:
- the introduction & abstract are wordy and boring. The discussion reads in a more straightforward way which emphasizes the "point" of the analysis. I would rewrite the intro & abstract to follow this type of model.
- While the authors are painstaking about chronicling all parts of their work and every detail, at some point it really is too much. I would cut down a number of the descriptions and explanations because they are too wordy. This can be achieved by thinking about what "really" is necessary to explain given previous work and standards in the field. Much of what they describe seems like a footnote (if even that much) in my opinion.
- The fonts in the supplementary file are too small to read. This should be worked out with the editor.
Overall, however, I believe the authors did a good job with this type of study and approached it and executed it in a reasonable fashion.
Author Response
Reviewer 2
My comments, which should be addressed before publication could be possible, are as follows:
- the introduction & abstract are wordy and boring. The discussion reads in a more straightforward way which emphasizes the "point" of the analysis. I would rewrite the intro & abstract to follow this type of model.
- Response: Thank you for your comments. We were limited by the complex question that is childhood obesity and the specific area within that covered by our work (i.e., the systems approach to prevention of obesity and the importance of the school setting). This led to explaining the issue and our chosen direction of work in the introduction section, including the introduction of the Cochrane review that served as our data source, and our study aim. All of this is covered in just 302 words, which we feel is acceptable. We have attempted to make the phrasing more straightforward, as advised. We have not edited the abstract as it already contains the bare minimum of essential information on methods and findings, making it shorter will lead to loss of important information.
- While the authors are painstaking about chronicling all parts of their work and every detail, at some point it really is too much. I would cut down a number of the descriptions and explanations because they are too wordy. This can be achieved by thinking about what "really" is necessary to explain given previous work and standards in the field. Much of what they describe seems like a footnote (if even that much) in my opinion.
Response: Thank you for your comment. This is somewhat intentional on our part. We have tried to move as much of the methods to supplementary information as possible. However, realist synthesis is relatively new, and the methods are still evolving. The existing guidance (e.g., RAMESES) is not always clear or prescriptive, with published realist reviews often skipping the detail in terms of implementing or operationalising the guidance. This is especially true for assessing rigour which is an important step to be carried out in the realist synthesis alongside relevance assessment. We have operationalised rigour in the best way we could in the absence of examples in published literature. We have tried to address this in our work and presented enough detail in the methods for someone to replicate our methods. We also believe and hope that our detailed presentation of methods will help discussions and evolve methods in the field. Finally, the methods section is just over 600 words long, which we believe to be reasonable.
- The fonts in the supplementary file are too small to read. This should be worked out with the editor.
Response: Thank you for pointing this out. The fonts unfortunately are small for the PDF of images presented for the development of programme theory. These can be read by zooming in to the images. We have tried to address this however have managed only to go so far without needing to cut image margins.
Overall, however, I believe the authors did a good job with this type of study and approached it and executed it in a reasonable fashion.
Response: Thank you.
Reviewer 3 Report
This paper is well structured, with a clear and detailed methodology, well-organised results, interesting discussion regarding other meta-analyses on the topic, and relevant considerations for further studies.
A robust dataset was used for analysing childhood obesity prevention interventions in recent years to understand why and how an intervention works. However, the period of paper analysis should be stated clearly in the Methodology.
Although stated at table footnotes, the first time the acronyms appear in the text, they should be explained. Therefore, please make these minor changes:
- Line 131: change “PA” to “physical activity (PA)”.
- Line 211: change “MVPA” to “moderate to vigorous physical activity physical activity (MVPA)”.
Author Response
Reviewer 3
This paper is well structured, with a clear and detailed methodology, well-organised results, interesting discussion regarding other meta-analyses on the topic, and relevant considerations for further studies.
Response: Thank you.
A robust dataset was used for analysing childhood obesity prevention interventions in recent years to understand why and how an intervention works. However, the period of paper analysis should be stated clearly in the Methodology.
Response: Thank you. We have now included the search span of the Cochrane 2019 review in the inclusion criteria section in line 78.
Although stated at table footnotes, the first time the acronyms appear in the text, they should be explained. Therefore, please make these minor changes:
- Line 131: change “PA” to “physical activity (PA)”.
- Line 211: change “MVPA” to “moderate to vigorous physical activity physical activity (MVPA)”.
Response: Thank you for pointing this out. We have corrected the error now.